

# Adjuvant chemoradiotherapy versus chemotherapy or radiotherapy in advanced endometrial cancer: a systematic review and meta-analysis

Hariyono Winarto[1], Naufal A. A. Ibrahim[2], Yan M. Putri[2], Faiqueen D. S. F. Adnan[2] and Eka D. Safitri[3]

[1] Gynecologic Oncology Division, Obstetrics and Gynecology Department, Dr. Cipto Mangunkusumo Hospital, Universitas Indonesia, Jakarta Pusat, Jakarta, Indonesia
[2] Faculty of Medicine, Universitas Indonesia, Jakarta Pusat, Jakarta, Indonesia
[3] The Center for Clinical Epidemiology and Evidence-Based Medicine, Dr. Cipto Mangunkusumo Hospital, Universitas Indonesia, Jakarta Pusat, Jakarta, Indonesia

## ABSTRACT

**Background:** Endometrial cancer is one of the most common gynecological cancer in the world. However, the available adjuvant therapies, chemotherapy (CT) and radiotherapy (RT), demonstrated several limitations when used alone. Therefore, we conducted a meta-analysis to investigate the clinical effectiveness of chemoradiotherapy (CRT) based on overall survival (OS) and disease-free survival (DFS).

**Methods:** A literature search was performed on five databases and one clinical trial registry to obtain all relevant articles. Search for studies was completed on September 9, 2021. A meta-analysis was conducted to determine the overall hazard ratio with the 95% Confidence Interval.

**Results:** A total of 17 articles with 23,975 patients in the CRT *vs* RT group and 50,502 patients in the CRT *vs* CT group were included. The OS Hazard Ratios (HR) of CRT compared to RT was 0.66 (95% CI [0.59–0.75]; $P < 0.00001$). Compared to CT, the OS HR was 0.70 (95% CI [0.64–0.78]; $P < 0.00001$). CRT also significantly improved the DFS compared to CT only (HR 0.79, 95% CI [0.64–0.97]; $P = 0.02$) However, CRT did not improve the DFS compared to RT only, with HR of 0.71 (95% CI [0.46–1.09]; $P = 0.12$).

**Conclusion:** Adjuvant CRT can significantly improve OS compared to CT or RT alone and improve the DFS compared to CT alone in patients with advanced endometrial cancer. Further research is needed to identify the optimal CRT regimen, and to whom CRT will be most beneficial.

Corresponding author
Hariyono Winarto,
hariyono.winarto@ui.ac.id

# INTRODUCTION

Accounts for 417,367 new cases and 97,370 deaths just in 2020, endometrial cancer is one of the most diagnosed gynecological cancer in the world (*World Health Organization,*

2020). Obesity, older age (≥55 years), exposure to estrogen, early menarche, and late menopause are among the risk factors (*World Health Organization, 2020*). While many other cancer incidences have decreased, endometrial cancer incidence has been rising by 21% from the past decade (*Sorosky, 2012*).

Since 1988, the International Federation of Gynecology and Obstetrics (FIGO) has decided to change the classification of endometrial cancer from clinical staging to surgical staging (*Creasman, 2009*). Most patients were initially presented to healthcare facilities with postmenopausal bleeding as primary complaint (*Stubert & Gerber, 2016*). Although it accounts for a smaller proportion, a higher FIGO stage is associated with poorer prognosis. The 5-year survival rate was around 30% to 60% in patients with advanced stages, in contrast to 80% to 97% in patients with early stages (*Kosary, 2007*).

The current treatment for advanced-stage endometrial cancer is surgery followed by adjuvant therapy. Through surgery, prognosis stratification and identification of appropriate adjuvant therapy was done. Adjuvant therapy is important to reduce the likelihood of cancer recurrence and increase overall survival (OS) (*Concin et al., 2021*). Radiotherapy (RT) has been used as adjuvant therapy for its ability to control local recurrence. Because RT demonstrated limited impact on distal recurrence, physicians have considered adjuvant chemotherapy (CT) (*DeLeon, Ammakkanavar & Matei, 2014*). However, CT alone has been shown to have a limited impact on preventing pelvic recurrence, which accounts for 18% as the first relapse site in the advanced stage (*Randall et al., 2006*). Currently, the clinical effectiveness of chemoradiotherapy (CRT) in advanced-stage endometrial cancer has not yet been established. Recently, several studies showed that CRT has a benefit on OS in advanced-stage endometrial cancer (*Lee & Viswanathan, 2012*; *Lester-Coll et al., 2016*; *Secord et al., 2013*). A meta-analysis on the effect of CRT on endometrial cancer has been previously done by *Park et al. (2013)*. However, the study focused more on high-risk endometrial cancer and only compared the effect of CRT to RT. The number of studies used in that meta-analysis for advanced stage endometrial cancer was relatively small. Since there have been several studies published after 2013, including two large clinical trials, an update on the effect of CRT to RT or CT in advanced stage is needed to confirm the benefit.

We conducted a systematic review and meta-analysis to determine CRT's clinical effectiveness compared to RT or CT only as adjuvant treatment for women with advanced-stage endometrial cancer. We evaluate the clinical effectiveness according to the OS and disease-free survival (DFS) of each treatment modality.

## MATERIALS AND METHODS

### Protocol and registration

A protocol was established before writing and registered in the International prospective register of systematic reviews (PROSPERO) on June 1, 2021 (CRD42021252529). We report this review according to the Preferred Reporting Items for Systematic Reviews and Meta-Analyses (PRISMA) checklist (*Moher et al., 2009*) (Table S1).

## Eligibility criteria

Studies were included if they fulfilled the following criteria: (a) the subject was advanced stage endometrial cancer patients who underwent surgery, (b) RCT or observational studies, (c) the study compared the result between CRT and CT or RT alone, and (d) the OS or DFS hazard ratio (HR) with 95% confidence interval (CI) and $p$-value were stated or can be calculated from the Kaplan-Meier Curve. The overall survival was defined as the time from the starting time point to the date of death and disease-free survival was defined as the time from the starting time point until the disease reoccurred. The starting time point for RCTs was randomization. For observational studies, the starting point was the date of diagnosis. However, due to limited number of studies, we also included studies that did not specify the starting time point. The HR can be calculated if the following data can be estimated from the curve: (1) event-free at the beginning, (2) censored and at risk throughout the interval, and (3) the number of events from every interval (*Tierney et al., 2007*). If the last criteria were not fulfilled, we inquire about the data through e-mail. The studies were included if the author had provided the required information. Studies were excluded if the full text is not available, or they were not written in English.

## Search strategy

We started the search for all published studies from electronic databases on April 10, 2021. We conducted preliminary search before PROSPERO registration to determine whether the volume of relevant studies is sufficient to conduct a systematic review and meta-analysis. Study search was completed on September 9, 2021. We searched for those studies from several databases, including SCOPUS, Medline (PubMed), EBSCO, Embase, Cochrane Central Register of Controlled Trials (CENTRAL) on the Cochrane Library, Web of Science, and ClinicalTrials.gov. We used key terms including "advanced stage endometrial cancer", "chemoradiotherapy", "chemotherapy", "radiotherapy", "outcome" and "survival" (Table S2). We also searched for additional studies through manual hand-searching and tracing of citations from related studies. Author NAAI, FDSA, and YM conducted the search for eligible studies.

## Study selection

All studies were exported to Mendeley software and duplicates were removed. Four reviewers (NAAI, FDSA, YM and HW) screened the titles and abstracts independently. The remaining articles will be assessed from their full text for their eligibility by the same three reviewers independently. Any disagreements will be resolved through discussion with a fifth reviewer (EDS).

## Quality assessment

The included studies were critically appraised by four independent reviewers (NAAI, FDSA, YM and HW) using the Cochrane Collaboration's tool for assessing the risk of bias in randomized trials for RCTs (*Higgins et al., 2011*) and the Newcastle Ottawa Scale (NOS) for retrospective studies (*Wells et al., 2000*). For RCTs, the risk of bias were assessed through selection, performance, detection, attribution, reporting, and other domains.

We then categorized the risk of bias as 'low risk,' 'high risk,' or 'unclear risk' of bias for each domain. For retrospective studies, total NOS score of 0 to 4 was categorized as low quality (high risk of bias), 5 to 7 as moderate quality (moderate risk of bias), and 8 to 9 as high quality (low risk of bias). Any discrepancy was resolved through discussion with a fifth reviewer (EDS) to reach an agreement.

## Data extraction and synthesis

We developed a data extraction form for this review. The data extracted included: first author, year of publication, study location and period, study design, the total number of enrolled subjects, baseline population characteristics (age and performance score), details of diagnosis (FIGO stage, tumor grade, types, and extension), the total number of intervention groups and details of intervention (modalities, dose, cycle length), risk of bias, follow up duration, and outcomes (OS and DFS). We presented the outcomes in the HR with the 95% CI. If the articles did not report the HR, we estimated the HR from the Kaplan-Meier curves using the method purposed by *Tierney et al. (2007)*. We contacted the author if there was missing or incomplete data.

## Statistical analysis

The quantitative data were exported to Review Manager 5.4 and pooled in a meta-analysis only if appropriate. We used the inverse variance method to obtain the pooled HR. Studies were considered to have moderate heterogeneity if I2 >30%, substantial heterogeneity if I2 >50%, and considerable heterogeneity if I2 >75% (*Higgins et al., 2021*). Sources of heterogeneity were assessed for any studies with substantial heterogeneity or more.
To detect any risk of publication bias, we constructed funnel plots in Review Manager 5.4 and performed Egger's Test in Stata 17. Symmetrical funnel plots indicated low risk of publication bias. The studies were considered to have potential risk of publication bias if *p*-value on Egger's Test is less than 0.05.

## Sensitivity analysis

We conducted a sensitivity analysis to identify the possible contribution of specific clinical or methodological differences between the included studies. Sensitivity analysis was performed by omitting studies with a high risk of bias leaving only studies with low risk of bias. Possible study characteristics that contribute to high risk of bias are inadequate follow-up period, different study type, small sample size, *etc*. Sensitivity analysis on specific treatment regimens was not conducted due to the lack of data.

# RESULTS

## Search selection

Randomized trials and retrospective studies that compared adjuvant CRT with CT or RT alone in advanced stage (stage III and IV) endometrial cancer were included. Our initial searches resulted in 293 articles and were reduced to 242 after duplicates were removed. Records screening for eligible studies yielded 26 articles. Following the assessment of full-text articles, nine articles were excluded due to unavailable HR and no comparison

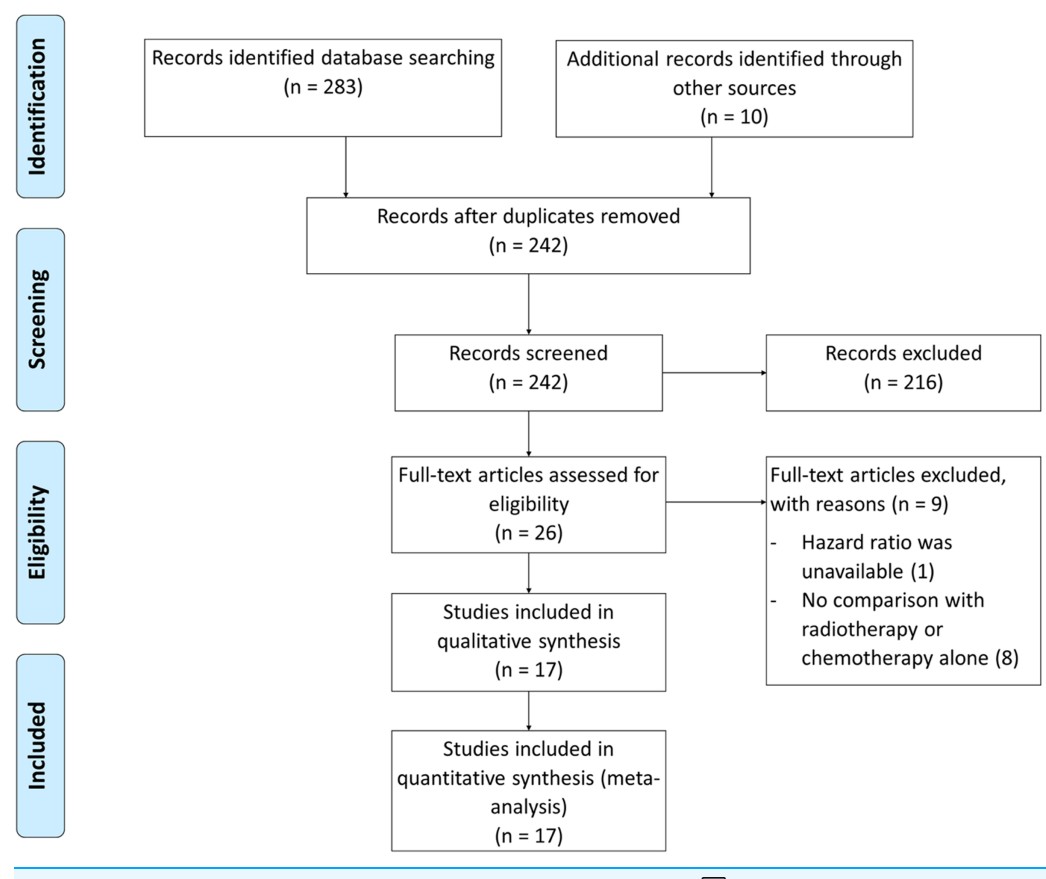

**Figure 1 Prisma diagram for study selection.**

with monotherapy. Seventeen articles were included, consisting of 2 RCT (*de Boer et al., 2019*; *Matei et al., 2019*) and 15 retrospective studies (*Lee & Viswanathan, 2012*; *Lester-Coll et al., 2016*; *Secord et al., 2013*; *Pichatechaiyoot et al., 2014*; *Secord et al., 2007*; *Tai et al., 2019*; *van Weelden et al., 2020*; *Albeesh et al., 2019*; *Lim et al., 2020*; *Kahramanoglu et al., 2019*; *Wong et al., 2016*; *Nakayama et al., 2010*; *Boothe et al., 2016*; *Goodman et al., 2019*; *Xiang, English & Kidd, 2019*; *Hogberg et al., 2010*) (Fig. 1).

## Characteristics of studies

Table 1 and Table S3 summarized the characteristics of included studies comparing adjuvant CRT with RT alone. Table 2 and Table S4 summarized the characteristics of included studies comparing adjuvant CRT with CT alone. Fourteen studies (*Lee & Viswanathan, 2012*; *Secord et al., 2013*; *de Boer et al., 2019*; *Pichatechaiyoot et al., 2014*; *Secord et al., 2007*; *Tai et al., 2019*; *van Weelden et al., 2020*; *Albeesh et al., 2019*; *Lim et al., 2020*; *Kahramanoglu et al., 2019*; *Wong et al., 2016*; *Nakayama et al., 2010*; *Boothe et al., 2016*; *Goodman et al., 2019*) compared adjuvant CRT with RT alone, and 13 studies (*Lester-Coll et al., 2016*; *Secord et al., 2013*; *Matei et al., 2019*; *Pichatechaiyoot et al., 2014*; *Secord et al., 2007*; *Tai et al., 2019*; *van Weelden et al., 2020*; *Kahramanoglu et al., 2019*; *Wong et al., 2016*; *Nakayama et al., 2010*; *Boothe et al., 2016*; *Goodman et al., 2019*; *Xiang, English & Kidd, 2019*) compared adjuvant CRT with CT. The two included RCT

**Table 1 Baseline characteristics of included studies comparing chemoradiotherapy and radiotherapy alone in advanced endometrial cancer.**

| First author (Year) | Study design (Location) | Median/ Mean follow-up period (Months) | Treatment groups | Sample (N) | Age (Years) Mean ± SD/ Median (IQR) | Non-endometrioid N (%) | Grade 3 N (%) | OS (%) | DFS (%) | HR (95% CI) | NOS |
|---|---|---|---|---|---|---|---|---|---|---|---|
| *de Boer et al. (2019)* | RCT (PORTEC-3) | 60.2 | CRT: cisplatin 2 cycles and EBRT, carboplatin + paclitaxel 4 cycles | 152 | – | – | – | 78.5 | – | 5-year OS: 0.63 [0.41–0.99], P = 0.043 | |
| | | | RT: EBRT48.6 gy | 143 | – | – | – | 68.5 | – | | |
| *Secord et al. (2007)* | ROS (United States) | 38 | CRT: platinum + paclitaxel + adriamycin (85%) and WPRT + extended field ± VBT | 83 | 63 (9.6) | 49 (59) | 62 (75) | 79.0 | 62.0 | 3-year OS: 0.49 [0.28–0.85], P = 0.012 3-year DFS: 0.62 [0.39–0.97], P = 0.037 | 6 |
| | | | RT: WAI (39%) or WPRT (35%) or WPRT and extended field ± VBT (12%) | 171 | 65 (10.8) | 92 (54) | 85 (50) | 70.0 | 59.0 | | |
| *Secord et al. (2013)* | ROS (United States) | 42 | CRT: paclitaxel + carboplatin or doxorubicin + cisplatin and WPRT ± extended filed ± VBT in sandwich (38%) or chemotherapy-RT (43%) regimen | 161 | 60 ± 10 | 59 (47) | 84 (52) | 90.0 | – | 3-year OS: 0.9 [0.28–3.33], P = 0.91 | 6 |
| | | | RT: WPRT ± extended filed ± VBT | 45 | 67 ± 12 | 25 (56) | 27 (63) | 95.0 | – | | |
| *Pichatechaiyoot et al. (2014)* | ROS (Thailand) | 19.9 | CRT: cisplatin + adriamycin (64%) in chemotherapy-RT regimen (55%) RT-chemotherapy regimen (18%) | 11 | 56.1 ± 12 | 1 (9) | 6 (55)* | 75.0 | 38.9 | 3-year OS: 0.86 [0.18–4], P = 0.85 3-year DFS: 0.66 [0.18–2.4], P = 0.74 | 6 |
| | | | RT: VBT or WPRT, VBT + WPRT | 33 | 57.2 ± 10. | 1 (3) | 21 (64) | 71.6 | 52.5 | | |
| *Lee & Viswanathan (2012)* | ROS (United States) | 48 | CRT: caboplatin + paclitaxel or paclitaxel + doxorubicin + cyclophosphamide and EBRT | 44 | 57.5 | 0 (0) | 19 (43) | 90.0 | 79.0 | 5-year OS: 0.20 [0.05–0.75], P = 0.02 5-year DFS: 0.12 [0.03–0.49], P = <0.01 | 6 |
| | | | RT: EBRT45 gy | 18 | 63 | 0 (0) | 3 (17) | 67.0 | 63.0 | | |
| *Albeesh et al. (2019)* | ROS (Canada) | 64 | CRT: carboplatin + paclitaxel 4-6 cycles and EBRT | 37 | 64 (31–83) | 19 (54) | 25 (37) | 61.0 | 51.0 | 5-year OS: 0.53 [0.28–1.00], P = 0.05 5-year DFS: 0.90 [0.45–1.79], P = 0.77 | 6 |
| | | | RT: EBRT45 gy in 1.8 gy per fraction | 67 | 68(42–89) | 17 (24) | 24(65) | 67.0 | 67.0 | | |
| *Tai et al. (2019)* | ROS (Taiwan) | 45.5 | CRT: one of two types of chemotherapy and RT in sandwich or sequential regimen | 28 | – | 0 (0) | 12 (43) | – | – | 5-year OS: 0.78 [0.25–2.37], P = 0.662 | 6 |
| | | | RT: EBRT5040cgy ± VBT | 87 | – | 0 (0) | 24 (28) | – | – | | |
| *Lim et al. (2020)* | ROS (Singapore) | 67.5 | CRT: carboplatin + paclitaxel 6 cycles (61%) and EBRT and VBT | 41 | 51 (34–66) | 0 (0) | 14 (34) | 89.3 | 87.2 | 5-year OS: 0.74 [1.18–2.90], P = 0.666 5-year DFS: 0.87 [0.26–2.94], P = 0.822 | 6 |
| | | | RT: EBRT45-50.4 gy and VBT 10 gy | 55 | 55 (30–85) | 0 (0) | 6 (11) | 77.2 | 75.9 | | |

| First author (Year) | Study design (Location) | Median/ Mean follow-up period (Months) | Treatment groups | Sample (N) | Age (Years) Mean ± SD/ Median (IQR) | Non-endometrioid N (%) | Grade 3 N (%) | OS (%) | DFS (%) | HR (95% CI) | NOS |
|---|---|---|---|---|---|---|---|---|---|---|---|
| *van Weelden et al. (2020)* | ROS (Netherland) | – | CRT: unspecified chemotherapy regimens and EBRT | 175 | 62.9 ± 9.5 | 85 (49) | 109 (62) | 61.0 | – | 5-year OS: 0.58 [0.44–0.77], $P = <0.001$ | 6 |
| | | | RT: EBRT | 650 | 66.5 ± 10.5 | 114 (18) | 262 (40) | 55.0 | – | | |
| *Kahramanoglu et al. (2019)* | ROS (Kazakhstan) | 42 | CRT: mostly carboplatin + paclitaxel + EBRT ± VBT | 544 | – | 0 (0) | – | 72.8 | 62.0 | 5-year OS: 1.1 [0.51–2.38] DFS: 1.02 [0.76–1.39] | 7 |
| | | | RT: EBRT ± VBT | 179 | – | 0 (0) | – | 77.1 | 67.0 | | |
| *Wong et al. (2016)* | ROS (United States) | 38.9 | CRT: unspecified chemotherapy regimens + EBRT or VBT | 2,522 | 60 | – | 990 (38) | 72.6 | – | 5-year OS: 0.77 [0.67–0.89] | 7 |
| | | | RT: EBRT alone or VBT or EBRT ± VBT | 1,265 | 64 | – | 416 (16) | 63.9 | – | | |
| *Nakayama et al. (2010)* | ROS (Japan) | 54 | CRT: unspecified (chemotherapy followed by RT (77%), RT followed by RT (23%)) | 26 | 54 ± 9.3 | 24 | 6 | 77 | – | 5-year OS: 0.21 [0.09–0.45], $P = 0.0521$ | 4 |
| | | | RT: WPRT (50%), WPRT + para-aortic RT (46%) | 20 | 56 ± 11.8 | 18 | 4 | 0 | – | | |
| *Boothe et al. (2016)* | ROS (United States) | 39 | CRT: unspecified sequential/ concurrent chemotherapy and RT regimen | 9,595 | – | 0 (0) | – | – | – | 5-year OS: 0.67 [0.64–0.71], $P = <0.01$ | 6 |
| | | | RT: EBRT, brachytherapy, or both | 4,486 | – | 0 (0) | – | – | – | | |
| *Goodman et al. (2019)* | ROS (United States) | – | CRT: unspecified regimen (RT before chemotherapy or RT after chemotherapy) | 2,070 | – | 0 (0) | 0 (0) | 78.1 | – | 5-year OS: 0.78 [0.54–1.13], $P = 0.18$ | 6 |
| | | | RT: unspecified RT regimen | 1,267 | – | 0 (0) | 0 (0) | 64.5 | – | | |

**Note:**
CI, Confidence Interval; CRT, Chemoradiotherapy; CT, Chemotherapy; DFS, Disease Free Survival; EBRT, External Beam Radiotherapy; HR, Hazard Ratio; IQR, Interquartile Range; NOS, Newcastle Ottawa Scale; OS, Overall Survival; RCT, Randomized Clinical Trial; ROS, Retrospective Observational Study; VBT, Vaginal Brachytherapy; WAI, Whole-Abdominal Irradiation; WPRT, Whole Pelvis Radiotherapy.

were the PORTEC-3 trial and the Gynecologic Oncology Group (GOG)—258 trials. The PORTEC-3 trial compared the OS between patients treated with CRT and RT only. The GOG-258 trial compared the DFS between patients treated with CRT and CT only. Although OS was one of the end points in the GOG-258 trial, the data were not adequately matured to allow comparison between groups. Most of the observational studies (*Lee & Viswanathan, 2012*; *Lester-Coll et al., 2016*; *Secord et al., 2013*; *Secord et al., 2007*; *Tai et al., 2019*; *van Weelden et al., 2020*; *Kahramanoglu et al., 2019*) used the date of diagnosis as their starting time point. *Secord et al. (2007)* and *Kahramanoglu et al. (2019)* counted the survival time after surgery, while *Pichatechaiyoot et al. (2014)* and *Goodman et al. (2019)* counted after the treatment completion. Four studies did not specify the starting time point. Treatment regimens for adjuvant CT and RT vary among all studies.

**Table 2 Baseline characteristics of included studies comparing chemoradiotherapy and chemotherapy alone in advanced endometrial cancer.**

| First author (Year) | Study design (Location) | Median/ Mean follow-up period (Months) | Treatment groups | Sample (N) | Age (years) Mean ± SD/Median (IQR) | Non-endometrioid N (%) | Grade 3 N (%) | OS (%) | DFS (%) | HR (95% CI) | NOS |
|---|---|---|---|---|---|---|---|---|---|---|---|
| Matei et al. (2019) | RCT (GOG 258) | 47 | CRT: cisplatin + EBRT + cisplatin | 294 | – | – | – | – | – | 5-year DFS: 1.02 [0.67–1.57], P = 0.92 | |
| | | | CT: cisplatin + paclitaxel | 275 | – | – | – | – | – | | |
| Secord et al. (2007) | ROS (United States) | 38 | CRT: platinum + paclitaxel + adriamycin (85%) and WPRT + extended field ± VBT | 83 | 63 (9.6) | 49 (59) | 62 (75) | 79.0 | 62.0 | 3-year OS: 0.62 [0.34–1.14], P = 0.122 3-year DFS: 0.64 [0.39–1.04], P = 0.076 | 6 |
| | | | CT: platinum + adriamycin + cyclophosphamide (33%) or platinum + paclitaxel + adriamycin (58%) | 102 | 67 (9.7) | 81(79) | 85(84) | 33.0 | 19.0 | | |
| Secord et al. (2013) | ROS (United States) | 42 | CRT: paclitaxel + carboplatin or doxorubicin + cisplatin and WPRT ± extended filed ± VBT in sandwich (38%) or ct-RT (43%) regimen | 161 | 60 ± 10 | 59 (36) | 84 (52) | 90.0 | – | 3-year OS: 0.25 [0.1–0.625], P = 0.004 | 6 |
| | | | carboplatin + paclitaxel or doxorubicin + paclitaxel or other regimens | 46 | 60 ± 11 | 14 (30) | 12 (27) | 78.0 | – | | |
| Pichatechaiyoot et al. (2014) | ROS (Thailand) | 19.9 | CRT: cisplatin + adriamycin (64%) in ct-RT regimen (55%) RT-CTregimen (18%) | 11 | 56.1 ± 12.0 | 1 (9) | 6 (55) | 75.0 | 38.9 | 3-year OS: 0.79 [0.11–5.55], P = 0.81 3-year DFS: 0.94 [0.21–4.35], P = 0.97 | 6 |
| | | | cisplatin + adriamycin (50%) or carboplatin + paclitaxel (30%) or cisplatin + cyclophosphamide (20%) | 10 | 54.3 ± 9.5 | 3 (30) | 6 (60) | 60.0 | 57.1 | | |
| Lester-Coll et al. (2016) | ROS (United States) | 59.6 | CRT: unspecified regimens | 3,479 | 61 (55–68) | 2,884 (70) | 1,953 (56) | 70.0 | – | 5-year OS: 0.63 [0.57–0.69], P = <0.001 | 7 |
| | | | CT: unspecified regimens | 6,358 | 62 (56–70) | 5,332 (76) | 4,270 (67) | 55.0 | – | | |
| Tai et al. (2019) | ROS (Taiwan) | 45.5 | CRT: one of two types of CT and RT in sandwich or sequential regimen | 28 | – | – | 12 (43) | – | – | 5-year OS: 0.49 [0.16–1.50], P = 0.191 | 6 |
| | | | CT: platinum + paclitaxel, anthracycline, cyclophosphamide, or ifosdamide | 79 | – | – | 31 (39) | – | – | | |
| van Weelden et al. (2020) | ROS (Netherland) | – | CRT: unspecified CT regimens and EBRT | 175 | 62.9 ± 9.5 | 85 (49) | 109 (62) | 61.0 | – | 5-year OS: 0.67 [0.49–0.92], P = 0.014 | 6 |
| | | | CT: unspecified CT regimens | 158 | 66.5 ± 10.5 | 128 (81) | 114 (72) | 39.0 | – | | |

| First author (Year) | Study design (Location) | Median/ Mean follow-up period (Months) | Treatment groups | Sample (N) | Age (years) Mean ± SD/Median (IQR) | Non-endometrioid N (%) | Grade 3 N (%) | OS (%) | DFS (%) | HR (95% CI) | NOS |
|---|---|---|---|---|---|---|---|---|---|---|---|
| *Kahramanoglu et al. (2019)* | ROS (Kazakhstan) | 42 | CRT: mostly carboplatin + paclitaxel + EBRT ± VBT | 544 | – | 0 (0) | – | 72.8 | 62.0 | 5-year OS: 0.87 [0.65–1.18] 5-year DFS: 0.75 [0.57–0.99] | 7 |
| | | | CT: mostly carboplatin + paclitaxel | 242 | – | 0 (0) | – | 69.8 | 53.7 | | |
| *Wong et al. (2016)* | ROS (United States) | 38.9 | CRT: unspecified CT + EBRT or VBT | 2,522 | 60 | – | 990 (37.6) | 72.6 | – | 5-year OS: 0.62 [0.55–0.71] | 7 |
| | | | CT: unspecified CT regimens | 1,533 | 61 | – | 678 (25.7) | 64.4 | – | | |
| *Xiang, English & Kidd (2019)* | ROS (United States) | 44.4 | CRT: unspecified CT regimens + EBRT | 5,311 | – | 1,765 (33) | – | 70 | – | 5-year OS: 0.83 [0.77–0.89] | 7 |
| | | | CT: unspecified CT regimens | 7,959 | – | 3,283 (41) | – | 62 | – | | |
| *Nakayama et al. (2010)* | ROS (Japan) | 54 | CRT: unspecified (CT followed by RT (77%), RT followed by RT (23%)) | 26 | 54 ± 9.3 | 24 | 6 | 77 | – | 5-year OS: 0.78 [0.43–1.41], $P = 0.0345$ | 4 |
| | | | CT: cisplatin + adriamycin + cyclophosphamide (73%), paclitaxel + carboplatin (13%) | 30 | 53 ± 9.7 | 28 | 8 | 60 | – | | |
| *Boothe et al. (2016)* | ROS (United States) | 39 | CRT: unspecified sequential/ concurrent regimen | 9,595 | – | 0 (0) | – | – | – | 5-year OS: 0.69 [0.66–0.72], $P = <0.01$ | 6 |
| | | | CT: unspecified CT regimen | 6,946 | – | 0 (0) | – | – | – | | |
| *Goodman et al. (2019)* | ROS (United States) | – | CRT: unspecified regimen | 2,070 | – | 0 (0) | 0 (0) | 78.1 | – | 5-year OS: 0.89 [0.63–1.26], $P = 0.5087$ | 6 |
| | | | CT: unspecified CT regimen | 2,465 | – | 0 (0) | 0 (0) | 68.9 | – | | |

**Note:**

CI, Confidence Interval; CRT, Chemoradiotherapy; CT, Chemotherapy; DFS, Disease Free Survival; EBRT, External Beam Radiotherapy; HR, Hazard Ratio; IQR, Interquartile Range; NOS, Newcastle Ottawa Scale; OS, Overall Survival; RCT, Randomized Clinical Trial; ROS, Retrospective Observational Study; VBT, Vaginal Brachytherapy; WPRT, Whole Pelvis Radiotherapy.

Platinum-based CT and external beam radiation therapy (EBRT) were used in most of the studies.

The total number of patients included was 23,975 patients in the CRT *vs* RT group and 50,502 patients in the CRT *vs* CT group. Five studies (*Lester-Coll et al., 2016*; *Wong et al., 2016*; *Boothe et al., 2016*; *Goodman et al., 2019*; *Xiang, English & Kidd, 2019*) were multi-center studies with samples more than 1,000 samples for each group. Most studies were conducted in the USA (*Lee & Viswanathan, 2012*; *Lester-Coll et al., 2016*; *Secord et al., 2013*; *de Boer et al., 2019*; *Matei et al., 2019*; *Secord et al., 2007*; *Wong et al., 2016*; *Boothe et al., 2016*; *Goodman et al., 2019*; *Xiang, English & Kidd, 2019*), five studies were conducted in Asia (*Pichatechaiyoot et al., 2014*; *Tai et al., 2019*; *Lim et al., 2020*; *Kahramanoglu et al., 2019*; *Nakayama et al., 2010*), one study was conducted in

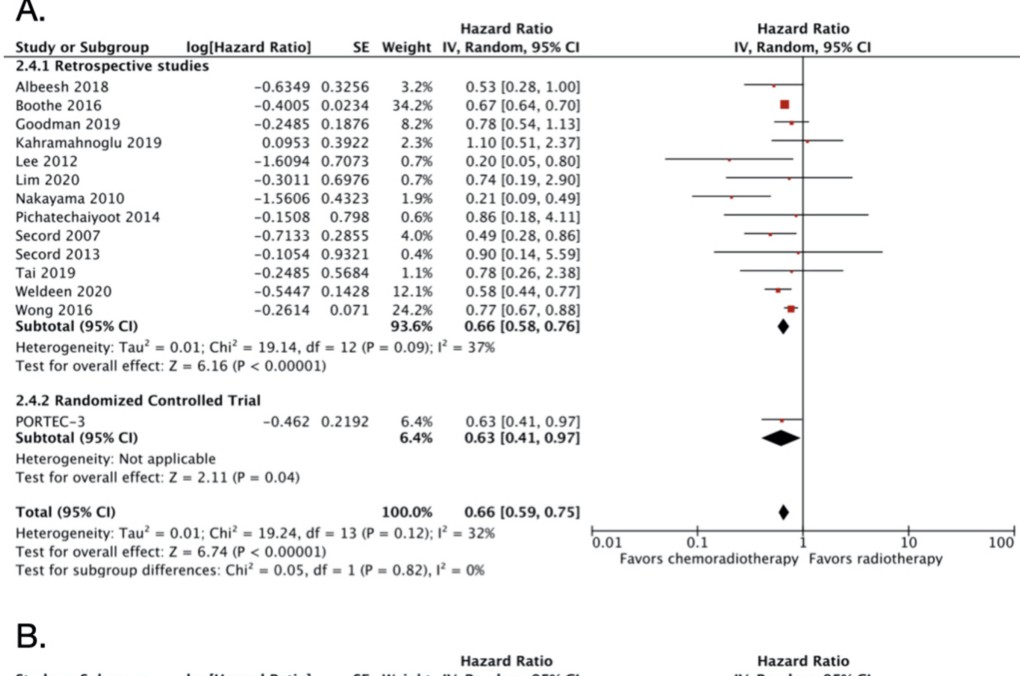

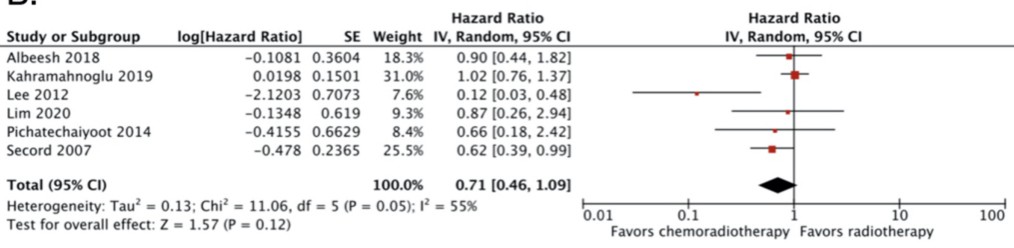

**Figure 2** Forest plot for (A) overall survival dan (B) disease-free survival, chemoradiotherapy *vs* radiotherapy alone in advanced endometrial cancer.    

Netherland (*van Weelden et al., 2020*), and one study was conducted in Canada (*Albeesh et al., 2019*).

The risk of bias assessment for the RCT studies (*de Boer et al., 2019*; *Matei et al., 2019*) resulted in low risk (Table S5). The NOS scores for the included retrospective studies (Table S6) were mostly 6 to 8, except for *Pichatechaiyoot et al. (2014)* and *Nakayama et al. (2010)* (5).

## CRT *vs* RT alone in advanced endometrial cancer

A total of 14 studies were included in this category, with 23,975 patients (Table 1). The pooled OS HR from all the included studies in this group was 0.66 (95% CI [0.59–0.75]; 23,975 patients; *P* < 0.00001; Fig. 2A). The result demonstrated moderate heterogeneity with I2 of 32% (*P* < 0.12). The PORTEC-3 trial (*de Boer et al., 2019*) was the only RCT in this comparison group. The funnel plot was relatively symmetrical, indicating low risk of publication bias (Fig. S1). The interpretation of the funnel plot was supported by the result from Egger's test which also resulted in low risks of publication bias (*P* value = 0.5465).

Sensitivity analysis was conducted by excluding studies with relatively small number of subjects (*Lee & Viswanathan, 2012*; *Secord et al., 2013*; *Pichatechaiyoot et al., 2014*; *Secord et al., 2007*; *Tai et al., 2019*; *Albeesh et al., 2019*; *Lim et al., 2020*; *Nakayama et al., 2010*). The adjusted OS HR after the exclusion was almost similar with HR of 0.69 (95% CI [0.69–0.76]) with no significant heterogeneity (I2 = 27%; $P < 0.23$; Fig. S2). Three studies obtained the HR from 3-years OS (*Secord et al., 2013*; *Pichatechaiyoot et al., 2014*; *Secord et al., 2007*). If the three studies were excluded, the OS HR became 0.67 (95% CI [0.58–0.76]; Fig. S3) with no heterogeneity (I2 = 44%; $P < 0.06$).

We also extracted DFS data from six studies with a total of 1,283 patients. The pooled DFS HR was 0.71 (95% CI [0.46–1.09]; 1,283 patients; Fig. 2B). Substantial heterogeneity was found with I2 of 55% ($P = 0.05$). Due to small number of included studies (less than 10 studies), we did not create a funnel plot for the DFS data. However, the Egger's test showed potential risks of publication bias ($P$ value = 0.0370).

### CRT *vs* CT alone in advanced endometrial cancer

A total of 13 studies were included in this category with 50,502 patients (Table 2). However, the one RCT included (GOG-258) (*Matei et al., 2019*) only reported DFS and not OS. The OS was improved significantly in patients with advanced stage endometrial cancer receiving adjuvant CRT compared to adjuvant CT only. The OS HR was 0.70 (95% CI [0.64–0.78]; 49,933 patients; $P < 0.00001$; Fig. 3A) with substantial heterogeneity (I2 = 71%; $P < 0.0001$). The funnel plot (Fig. S4) and the Egger's test result ($P$ value = 0.4686) indicated low risk of publication bias. Five studies enrolled less than 100 subjects for each treatment groups (*Secord et al., 2013*; *Pichatechaiyoot et al., 2014*; *Secord et al., 2007*; *Tai et al., 2019*; *Nakayama et al., 2010*). Because of the relatively small numbers of subjects enrolled, the result of those studies might not be representative. Therefore, both studies were excluded in the sensitivity analysis. After the exclusion, the adjusted OS HR was still similar with an increase in heterogeneity (0.71; 95% CI [0.64–0.79]; I2 = 81%; $P < 0.0001$; Fig. S5). Furthermore, three studies (*Secord et al., 2013*; *Pichatechaiyoot et al., 2014*; *Secord et al., 2007*) that reported 3-years OS were also excluded. After the exclusion of the three studies, the OS HR became 0.71 (95% CI [0.65–0.79]; Fig. S6) and the heterogeneity remained high (I2 = 76%; $P < 0.0001$).

The DFS data were obtained from four studies with a total of 1,561 patients. The pooled DFS HR was 0.79 (95% CI [0.64–0.97]; 1,561 patients; Fig. 3B) with no heterogeneity found (I2 = 0%; $P = 0.51$). The funnel plot for DFS data was also not created due to small number studies. The Egger's test resulted in low risk of publication bias ($P$ value = 0.7670).

## DISCUSSION

### Summary of main results

The included studies showed different significant confounding factors, however multivariate analysis of the HR was provided in all studies except for *Pichatechaiyoot et al. (2014)* and *Nakayama et al. (2010)*. Multivariate analysis from the other studies allowed correction for other covariates, therefore provide more accurate correlation between the adjuvant therapy and the outcome.

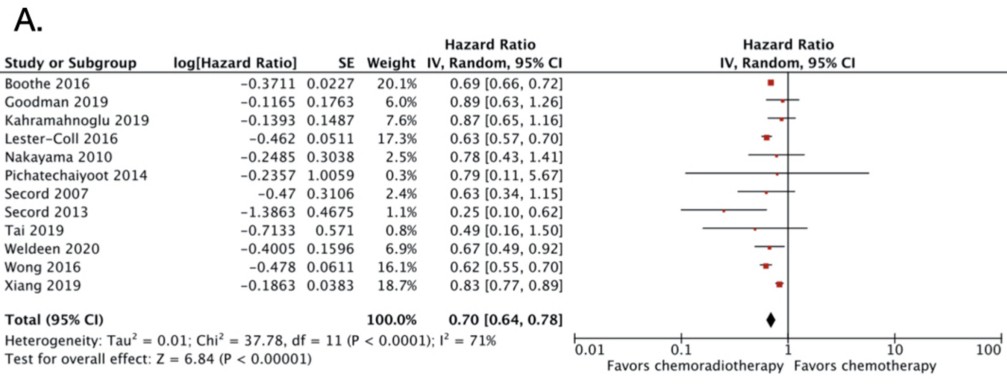

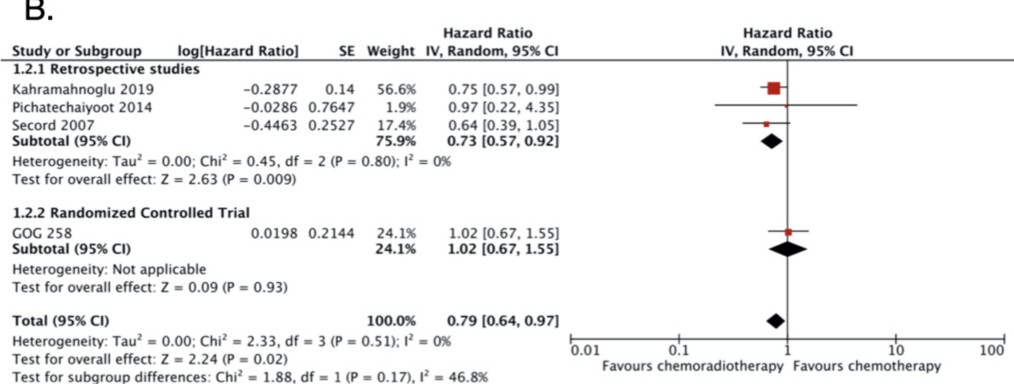

**Figure 3 Forest plot for (A) overall survival and (B) disease-free survival, chemoradiotherapy *vs* chemotherapy alone in advanced endometrial cancer.**

The result (Figs. 2A and 3A) implicates that a combination of adjuvant CT and RT improves the OS of women with advanced-stage endometrial carcinoma. However, only DFS of CRT *vs* CT showed favor to CRT (Fig. 3B), whereas DFS of CRT *vs* RT showed no statistical significance, even though the pooled HR point toward CRT (Fig. 2B).

It is important to note that there was a discordant result on DFS of CRT *vs* CT from GOG-258 and included retrospective studies (Fig. 3B). Although, ultimately pooled DFS favors CRT. Contributing factors to this difference might be firstly in GOG-258, both stage III and IV were included in the study. Meanwhile, only one retrospective study (*Pichatechaiyoot et al., 2014*) that included stage IV patients. In which, higher disease stage is known to be associated with worse disease prognosis. Second, GOG-258 carboplatin and paclitaxel combination was use as CT regimen in monotherapy, and an addition of cisplatin in CRT. Meanwhile, in the included retrospective studies other combination of CT regimens was used, although was not specified.

## Agreements with other studies

*Hogberg et al. (2010)* conducted two RCTs that included MaNGO ILIADE-III. MaNGO ILIADE-III aimed to compare the outcome of women who received RT and CRT in patients with stage IIB to stage III with a total of 156 subjects (*Lee & Viswanathan, 2012*). In accordance with our study, the result also showed that adjuvant CRT had better OS than

women who received RT only. However, this outcome was not statistically significant (HR 0.64, 95% CI [0.36–1.52]; $P = 0.07$). One of the possible explanations was the small number of subjects in the trial.

Another meta-analysis, *Park et al. (2013)*, also included a combination of observational studies and RCTs. Compared to RT only, the study concluded that CRT had a significant effect on the OS of advanced-stage endometrial cancer (OS HR 0.53, 95% CI [0.36–0.80]) (*Lester-Coll et al., 2016*). This conclusion supports the result of our study and provides more reliable evidence to implement the use of combination adjuvant therapy after surgery in women with advanced-stage endometrial cancer. Therefore, the conclusions from the studies above further support the result of our review.

## Overall completeness and quality of evidence

The sensitivity analysis result was consistent and did not differ from overall CRT *vs* RT and CRT *vs* CT group analyses. In which all result favors combination therapy rather than monotherapy. The overall risk of bias of the included studies was low to moderate. The result of our analysis was consistent with other studies, resulting in no to moderate heterogeneity.

We considered this result applicable to most patients with advanced-stage endometrial cancer. The survival benefit of CRT may vary depending on additional risk factors, such as age, tumor grade, and histological types. However, we were not able to perform subgroup analysis according to those risk factors due to insufficient data. Most studies did not include the performance status of their patients before receiving treatments on their data. The RCT study (*de Boer et al., 2019*; *Matei et al., 2019*) only included patients with a performance score of 0–2. It is important to note because not all patients with advanced disease have a high-performance score and patients with low-performance status are more susceptible to CT toxicity (*Azam et al., 2019*; *Sargent et al., 2009*). Further research is necessary to determine to whom CRT will be most beneficial and well-tolerated.

## Biological plausibility

In advanced stage endometrial cancer, where the cancer cells have spread outside the uterus, some physicians have considered combining CT and RT to improve OS. The hypothesis was that radiation therapy damages DNA and creates genomic instability leading to cellular death (*Galluzzi et al., 2007*). However, this cytotoxic effect exerts only in the area treated with radiotherapy without influencing disease progression outside the treated area. Adding CT reduces the risk of relapse on pelvic and other occult metastatic as it exerts cytoreductive and cytostatic effects systemically (*Schwandt et al., 2011*).

## Applicability of evidence and implication for practice

As reported in the forest plot, OS favors combination therapy. However, it is not the only important parameter regarding cancer patient's treatment. Hence, we also calculated the DFS, which more reflects the patient's quality of life compared to just OS.

We can conclude from the result before that the DFS of CRT *vs* RT yielded in no statistical difference, unlike CRT *vs* CT. This result showed that the role of RT as an

adjuvant treatment for advanced stage endometrial cancer is valuable. Previous studies have shown that radiotherapy lowers both the rate of local and regional recurrence (*Mundt et al., 2001*; *Klopp et al., 2009*). However, the availability of RT centers in some South-East Asian countries are much lower compared to the United States and European countries. For instance, in Indonesia, there are only 33 RT centers. Which means that on average, there is one RT center per 7.4 million of the Indonesian population. The low number of RT specialists has also been an issue for years, as cancer incidence continues to increase (*Hiswara, 2017*). The scarcity of RT treatment in many countries should be addressed, as we found in this study that RT alone is comparable to combination therapy in terms of quality of life.

### Limitation and potential bias in the review process

The result of OS in CRT *vs* CT yielded moderate heterogeneity (I2 = 71%). In addition, we combined the results of the RCT study with observational studies into one forest plot and only included observational studies for DFS result of CRT *vs* RT. Therefore, the result has lower level of evidence credibility compared to if we included RCTs. Conducting future trials regarding this matter are essential to improve our current knowledge.

In this review, funnel plot was not created for DFS result due to the inadequate number of studies (<10 studies) as suggested by *Ioannidis & Trikalinos (2007)*. However, the results of Egger-test indicate low impact of small-study effect except for DFS of CRT *vs* RT. Contributing factors might be due to the small number of included studies (six studies) and the type of study which only comprised of observational studies. Retrospective observational studies in general have lower level of evidence compared to RCT. Observational study is not randomized and not blinded. Also, in retrospective study, researcher formulated study hypothesis before data collection. Hence, these factors contribute to the higher level of bias and lower level of evidence compared to RCT. We suggest that to obtain higher certainty of evidence, more RCT should be conducted comparing monotherapy and CRT. In addition, subgroup analysis on specific treatment regimens might also be beneficial.

## CONCLUSIONS

Given the limitations of monotherapy, adjuvant CRT becomes a reasonable treatment option in advanced-stage endometrial cancer. There is a moderate quality of evidence with low risk of bias that adjuvant CRT can significantly improve OS compared to CT or RT alone in patients with advanced endometrial cancer. Further research is needed to identify the optimal CRT regimen and whom CRT will most benefit regarding toxicity and quality of life.

### Funding

The authors received no funding for this work.

## Competing Interests

The authors declare that they have no competing interests.

## Author Contributions

- Hariyono Winarto conceived and designed the experiments, authored or reviewed drafts of the article, and approved the final draft.
- Naufal A. A. Ibrahim conceived and designed the experiments, performed the experiments, analyzed the data, prepared figures and/or tables, and approved the final draft.
- Yan M. Putri conceived and designed the experiments, performed the experiments, analyzed the data, prepared figures and/or tables, and approved the final draft.
- Faiqueen D. S. F. Adnan conceived and designed the experiments, performed the experiments, analyzed the data, prepared figures and/or tables, and approved the final draft.
- Eka D. Safitri conceived and designed the experiments, authored or reviewed drafts of the article, and approved the final draft.

## Data Availability

The PRISMA checklist, search strategy for databases, characteristics of included studies, risk of bias assessment, funnel plot of overall survival, and sensitivity analysis are available in the Supplemental Files.

## Supplemental Information

Supplemental information for this article can be found online at http://dx.doi.org/10.7717/peerj.14420#supplemental-information.

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
