# Peer review of "Adjuvant chemoradiotherapy versus chemotherapy or radiotherapy in advanced endometrial cancer: a systematic review and meta-analysis"

_PeerJ, doi:10.7717/peerj.14420_

## Round 0.1 · original submission · Minor Revisions

Please carefully consider the reviewers’ suggestions before resubmitting.

Reviewer 1 ·

Basic reporting

The manuscript is clearly written and easy to read and follow. The article structure is consistent with the typical systematic literature review and meta-analysis publications. The background and rationale of this manuscript is by and large clear and sufficient. One specific comment below.

1. The authors mentioned there was a meta-analysis (Park 2013) that evaluated the benefit of adjuvant chemotherapy combined with radiotherapy for endometrial cancer. Suggest the authors to mention it in the introduction and clearly point out the difference of their work vs. Park 2013 and their additional contribution to the literature.

Experimental design

The design of systematic literature review and search criteria are by and large defined clearly. The statistical methods used to evaluate the survival outcomes (DFS and OS) are standard methods used in similar settings. Specific comments on search criteria are listed below.


1. The meta-analysis was designed to investigate the clinical effectiveness of CRT based on OS and DFS, but in the Eligibility Criteria section (line 77-84), the DFS HR was not included in the criteria. Please make clear in the manuscript if the studies that only included DFS HR without OS HR were included in the meta-analysis and explain the reason if not.

2. Definition of endpoints: In the data extraction table (Table 1), the authors did not show the definition of endpoints (DFS and OS). In many studies, the endpoint definitions (e.g., starting time point, specific events, etc.) are different, particularly for DFS, which may include different starting time points and types of events (relapse, recurrence, progression, etc.). Thus, it’s important to compare the endpoint definitions and make sure that the studies included in one analysis have comparable definitions. Please add and comment on the definitions of outcomes from the studies included in the meta-analysis.

3. For the criteria (d): the OS HR with 95% Ci and p-value were stated or can be calculated. Does “can be calculated” mean that the K-M curves of DFS and/or OS are available in the publication? If yes, please specify it clearly.

4. Did the authors search the conference abstracts, for example, ASCO and ESMO in the last 3 years? Some latest clinical trial results may be just presented at the recent medical conferences. If the authors only search full-text publications, the results in the conference abstracts may be missed.

5. Please add study follow-up time in the extraction table (Table 1). It’s an important variable to examine the comparability of studies and interpret the results.

Validity of the findings

The interpretation of results is correct and the conclusions are well stated. The discussion section also highlighted the major limitation of the study. My major comment on the validity of the findings in this manuscript is that the level of evidence credibility of this meta-analysis is not high given the limited randomized clinical trials included in the meta-analysis, but it's mainly due to limited randomized trials available. I have one specific comment on the sensitivity analysis in the manuscript.

1. In the sensitivity analysis, did authors examine the impact of specific treatment regimens on the pooled HR of DFS or OS? For example, platinum-based CT vs. other CT, or EBRT vs. radiation therapy?

Reviewer 2 ·

Basic reporting

Thank you for the invitation for reviewing the manuscript. The study examined the effect of chemoradiotherapy versus radiotherapy and chemotherapy alone in improving the overall survival and disease-free survival among patients with advanced endometrial cancer. This is a well-focused study with a smooth flow of information during the explanation of every step. Professional English is used throughout. Overall, the manuscripts meets the required standards for publication after minor edits.

Experimental design

The research question is focused, relevant & meaningful. The study approaches are valid.

Validity of the findings

All materials mentioned in manuscript are provided, and they are robust, statistically sound and clearly stated.

Additional comments

- Consider adding discussion on potential reasons/explanation for results that are not consistent with other studies, especially the discordant conclusion on DFS from randomized clinical trial and retrospective studies, chemoradiotherapy versus chemotherapy alone in advanced endometrial cancer.
- Consider adding discussion on possible biases, strength and weaknesses of clinical trials/observational studies included in this meta-analysis or in general. These discussion could provide guidance in the design of future studies.
- Are any criteria of publication year used during literature search? If yes, please state in manuscript.
- Clarify any funding source if available and if any potential conflict of interest identified.

---

## Round 0.2 · accepted · Accept

We are glad that you considered our journal for your valuable contribution.

Reviewer 1 ·

Basic reporting

The authors have addressed my comments.

Experimental design

The authors have addressed my comments.

Validity of the findings

The authors have addressed my comments.

Reviewer 2 ·

Basic reporting

All my comments have been addressed. I don't have further comments.

Experimental design

All my comments have been addressed. I don't have further comments.

Validity of the findings

All my comments have been addressed. I don't have further comments.

Additional comments

All my comments have been addressed. I don't have further comments.